# Cardiorespiratory Coordination in Collegiate Rowing: A Network Approach to Cardiorespiratory Exercise Testing

**DOI:** 10.3390/ijerph192013250

**Published:** 2022-10-14

**Authors:** Zacharias Papadakis, Michelle Etchebaster, Sergi Garcia-Retortillo

**Affiliations:** 1Human Performance Laboratory, Department of Health Promotion and Clinical Practice, College of Health and Wellness, Barry University, Miami Shores, FL 33161, USA; 2Department of Health and Exercise Science, Wake Forest University, Winston-Salem, NC 27109, USA; 3Complex Systems in Sport Research Group, Institut Nacional d’Educació Física de Catalunya (INEFC) University of Barcelona, 08007 Barcelona, Spain

**Keywords:** complex adaptive systems, coordinative variables, dynamic networks, network physiology, intra-individual co-variability, dynamic couplings, principal component analysis, time-series analysis, cardiovascular system, athlete’s performance evaluation

## Abstract

Collegiate rowing performance is often assessed by a cardiopulmonary exercise test (CPET). Rowers’ on-water performance involves non-linear dynamic interactions and synergetic reconfigurations of the cardiorespiratory system. Cardiorespiratory coordination (CRC) method measures the co-variation among cardiorespiratory variables. Novice (*n* = 9) vs. Intermediate (*n* = 9) rowers’ CRC (H_0_: Novice CRC = Intermediate CRC; H_A_: Novice CRC < Intermediate CRC) was evaluated through principal components analysis (PCA). A female NCAA Division II team (*N* = 18) grouped based on their off-water performance on 6000 m time trial. Rowers completed a customized CPET to exhaustion and a variety of cardiorespiratory values were recorded. The number of principal components (PCs) and respective PC eigenvalues per group were computed on SPSS vs28. Intermediate (77%) and Novice (33%) groups showed one PC_1_. Novice group formed an added PC_2_ due to the shift of expired fraction of oxygen or, alternatively, heart rate/ventilation, from the PC_1_ cluster of examined variables. Intermediate rowers presented a higher degree of CRC, possible due to their increased ability to utilize the bicarbonate buffering system during the CPET. CRC may be an alternative measure to assess aerobic fitness providing insights to the complex cardiorespiratory interactions involved in rowing during a CPET.

## 1. Introduction

Cardiopulmonary exercise testing (CPET) and the related concept of maximal oxygen consumption (VO_2max_) is considered as the most important indicator of endurance capacity, cardiorespiratory fitness, and health in sports science [1,2,3,4,5,6,7,8]. It measures a variety of variables (i.e., ventilation, VE; heart rate, HR; oxygen saturation, SpO_2_; ventilatory threshold, VT; expired fraction of oxygen, FeO_2_; expired fraction of carbon dioxide, FeCO_2_) linked to metabolic, cardiovascular and pulmonary responses during the CPET [7,9]. According to this concept, the human body has a limited capacity to utilize oxygen for muscle work as demonstrated by a plateau in VO_2max_ that indicates a physiological ceiling in cardiorespiratory capacity [10,11,12]. Since VO_2max_ is an important outcome for physical performance, a variety of tests designed to access VO_2max_ using a treadmill or cycle ergometer have been used, with the most popular ones to be the ramp and graded maximal incremental exercise tests [13,14,15,16].

Rowing as a sport has different biomechanical characteristics compared to running and cycling, so there are few rowing-specific VO_2max_ tests using work rate as a key element for the CPET setup on a Concept II rowing ergometer (i.e., stroke rate, critical velocity, power output) [17,18,19,20]. The most popular protocol to assess off-water VO_2max_ in rowers has been the simulation of the 2000 m distance time trial with maximal individual intensity with no differences observed compared to incremental exercise testing until exhaustion [19,20,21,22,23,24,25]. Competitive collegiate rowing requires a high physical demand [26], with metabolic contribution to 70–88% aerobic and 12–30% anaerobic [22,27]. Due to the high contribution of the aerobic system in rowing performance, direct VO_2max_ measurement has been shown to be highly correlated (e.g., r = 0.85–0.88) with actual rowing performance and to be the most important physiological determinant of rowing performance [20,27,28,29]. Therefore, the gold standard to assess direct off-water VO_2max_ in rowing is through gas analysis in a rowing ergometer (e.g., Concept II). Such off-water VO_2max_ test have reported maximum values of more of 6 L of oxygen per minute (L/min) or values close to 65 millimeters per kilogram of body weight per minute (mL/kg/min) [18,30,31,32,33,34,35,36,37,38,39,40], while collegiate female rowers have reported VO_2max_ values of 58 to 65 mL/kg/min [41]. Being able to measure and track the physical demands of rowers is crucial to both training and success in rowing [26]. Moreover, even though is difficult to ensure a controlled load output on rowing ergometers during a CPET [42], such a CPET allows for observing and describing both external (i.e., maximal power output) [34,42,43,44] and internal loads (i.e., VO_2max_, HR) [45,46,47,48,49].

The inability of the most commonly VO_2max_ used tests to capture maximum performances is due to their inability to capture the non-linear dynamic interactions among different physiological systems [50,51]. It has been proposed that the synergism of the cardiovascular and respiratory systems during a sub maximum or maximum exercise stimulus cannot be captured by the traditional VO_2max_ tests [52]. The framework of Network Physiology and, more specifically, Network Physiology of Exercise (NPE) [51,53,54,55,56,57,58] utilizes non-linear modeling and time series analysis of coordinative variables to investigate how different physiological systems coordinate and synchronize as a network [59,60,61,62]. For example, cardiorespiratory coordination (CRC) has been recently proposed as an alternative method to assess the synergism between the cardiorespiratory variables during an exercise stimulus [52,63,64,65,66,67]. This method requires the use of principal component statistical analysis (PCA) performed on time series cardiorespiratory data. Based on this statistical approach, the time synchronization as derived from different physiological systems reflects their shared co-variation between the involved physiological parameters, which at the end is represented through fewer principal components (PCs). The respective PCs are extracted in a decreasing fashion and represent the maximum covariance between the examined physiological variables, with the total number of PCs to represent the coordination level among them [67]. It is stated that the decrease in PCs and/or the increase PC eigenvalues reflects a higher level of cardiorespiratory efficiency-coordination [52,68].

Along this line of thought, recent work from our lab analyzed postprandial network interactions between autonomic nervous system and lipemia data in response to acute partial sleep deprivation and high-intensity interval exercise under the NPE framework. Even though we did not perform a true network analysis due to a non-time series data collection methodology, we reported that negative links in short-sleep high-intensity interval exercise (HIIE) condition reflected the influence of sleep on both the autonomic regulation and lipemia [69]. A time series analysis would have allowed us to both capture the synergistic reconfiguration of the autonomic nervous system and cardiometabolic lipemia and identify the causality between the physiological signals by creating a physiological postprandial network [59,66,70]. In contrast, when the same research question was analyzed using influential statistics, HIIE was cardioprotective regardless the impact of the short sleep [71,72].

It is clear that a research question analyzed under different frameworks may lead to different conclusions. Therefore, analyzing cardiorespiratory testing under the NPE framework may reveal the non-liner dynamic interactions of the cardiorespiratory system. Evidence supports the notion that CRC may be implicated to higher training adaptations [52] and training load [64] than VO_2max_ and other markers of aerobic fitness, pointing out CRC’s potential to substitute the traditional markers of aerobic capacity. Although metabolic, cardiovascular, and pulmonary systems work in coordination during a CPET, there is scarce evidence related to CRC in collegiate rowing CPET.

Therefore, this study investigated the measurement of CRC through PCA in collegiate rowers. We hypothesized that Intermediate rowers compared to Novice rowers will have higher CRC values. Authors seek to offer an alternative and more meaningful interpretation of rowers’ performance and training for maximal adaptation and prevent undertraining and overtraining. Results from this study may enrich the information provided by the traditional rowing ergometry tests to assess VO_2max_ [73,74,75,76].

## 2. Materials and Methods

### 2.1. Study Design and Participants

A cross-sectional observational study was conducted on a female rowing team that competes in NCAA Division II, as part of their annual pre-season physiological measurements. Rowers performed a preliminary rowing test to establish the initial rowing power for the customized discontinuous incremental rowing test to exhaustion. Based on the preliminary testing results, rowers were divided to Novice and Intermediate ones (Figure 1). Athletes’ direct off-water VO_2max_ capacity was measured through gas analysis using a Concept II rowing ergometer and a discontinuous incremental rowing protocol [77,78]. Besides measuring their VO_2max_ capacity, demographics and anthropometrics were also recorded.

As part of team ‘s requirements, all rowers (*n* = 18, age = 20.17 ± 2.28 SD years) supplied their consent to have both their VO_2max_ capacity and body composition assessed. All testing was performed on a single day by the same research personnel during morning hours in an air-conditioned levorotatory environment when no team practice was scheduled for 24 h prior to testing. The study was approved by the Ethics Committee of Barry University’s institutional review board #1851725-2 based on established policies on classroom and student research.

### 2.2. Procedures

#### 2.2.1. Anthropometrics and Body Composition

Height and weight were determined using an electronic scale and stadiometer (Seca 703), with participants removing their shoes prior to stepping on the scale [79]. Body composition was measured via a bipolar digital bioimpedance system with tatcile poles (OMRON Body Fat Analyzer, HBF-306BL, Omron Healthcare Corporation, Kyoto, Japan) and body fat percentage (%) was calculcated following standard procedures as previously described. Briefly, participants visited lab in fasting condition, with no eating and consuming water 2 h before testing and abstain from exercise 24 h prior to testing. All measurements happened during noon time, before lunch, and approximately 20 min prior to the cardiopulomonary exercise testing [79,80].

#### 2.2.2. Preliminary Customized Discontinuous Incremental Rowing Protocol—Initial Rowing Power

A customized discontinuous incremental rowing protocol based on rowers’ 60 s rowing speed was employed. According to this, rowers had to perform a rowing sprint of 60 s as fast as possible they could on a Concept II rowing ergometer to determine the initial power output for the actual cardiopulmonary exercise test (CPET). A priori power output of 250 Watts for at least 10 strokes in 60 s was set as cutoff point in order to classify the rowers into the Intermediate (>250 W) or to Novice (<250 W) group [77,78].

There is no clear consensus in the literature on Novice/Freshman rowers or DII rowers regarding the testing protocols and how to establish the starting wattage [77,78]. Due to this, we used a practical field approach to determine the starting power output. It is a widespread practice in rowing coaches to base the starting Wattage output on the performance of the 1-min all-out test. According to this common field practice, if a rower could maintain a power output of ~250 Watts for 10 to 15 strokes, then it was expected to at least make it to the fifth stage of the test and these values to represent a realistic oxygen consumption. However, if a rower could not achieve this initial output and strokes rate, then it was assumed that the lack of power is the reason why the test was terminated, without achieving a realistic maximum oxygen consumption [26,36,41,81,82].

#### 2.2.3. Cardiopulmonary Exercise Testing (CPET) Protocol

Having set the initial power, rowers completed a customized discontinuous incremental rowing test to exhaustion. Prior to the customized CPET, a warmup of 5 min in the Concept II rowing ergometer was performed. Participants were instructed to perform 2 min of easy rowing at a power output of <70 Watts, and the intensity was increased every minute as follows: 1 min between 70–100 Watts, 1 min between 100–130 Watts, and last minute was divided in 2 intervals of 30 s in which the power output was 130–160 Watts and >160 Watts, respectively. Following the warmup, the incremental discontinued protocol was performed. The incremental stages were set at 30 Watts for all rowers, while Intermediate rowers started at 70 Watts and the advanced ones at 100 Watts [77,78]. The duration of each stage was 2 min with 30 s rest in between. Since this was a maximum test until volitional fatigue exhaustion, rowers were expected to give their absolute best. Participants were encouraged to complete a maximum of 7 stages or until required power output could not be maintained. Following the CPET a cool down period of 3 min was performed, where participants were instructed to continue rowing at a power output of 50–70 Watts at their preferred stroke rate with heart rates to be below 100 beats per minute.

#### 2.2.4. Principal Components Analysis (PCA)

To analyze the CRC for each participant, we performed a PCA on the data series of the following selected cardiorespiratory variables: VE, FeO_2_, FeCO_2_, and HR. We excluded from the analysis VEqO_2_, VEqCO_2_, O_2_ pulse, RER, VO_2_, etc., due to their known deterministic mathematical relation (linear combination) with the selected variables [52]. There is diverse evidence about the use of dimensionality reduction by PCA in small samples, which indicates certain robustness in the estimates of shared variance that [83] pointed out some time ago. In this sense, it should be noted that the estimates in small samples should be more descriptive than inferential considerations, but appropriate to our objectives [84]. Continuous blood pressure monitoring could not be provided in this study. However, non-published results of our lab have shown similar results while analyzing CRC with and without continuous blood pressure measurement. 

### 2.3. Statistical Analysis

To analyze the suitability of the PCA implementation, we calculated Bartlett’s test for sphericity and the Kaiser-Mayer-Olkin (KMO) test for all participants. We determined the number of PCs using the Kaiser-Gutmann criterion and thus considered PCs with eigenvalues λ ≥ 1.00 as significant [85]. Given that the first PC (PC_1_) always contains the highest proportion of the data variance [63,66], the PC_1_ eigenvalues were compared between Nand Intermediate rowers by means Mann–Whitney U test. Effect size (Cohen’s d) was calculated when possible, to demonstrate the magnitude of standardized mean differences [86,87,88].

## 3. Results

Demographic characteristics are presented in Table 1.

The Bartlett’s sphericity test (*p* < 0.001) and the KMO index showed an acceptable sampling adequacy in both Novice (M = 0.61; SD = 0.12) and Intermediate rowers (M = 0.59; SD = 0.08). While 7 participants (77%) in the Intermediate group showed one PC, only 3 participants (33%) in the Novice group displayed 1 single PC (Table 2). The formation of an additional PC (i.e., PC_2_) in Novice rowers was the result of the shift of FeO_2_ or, alternatively, HR and VE, from the PC_1_ cluster of variables. As shown in Table 2, FeO_2_, VE and HR for Novice rowers showed the lowest projections onto PC_1_. Remarkably, eigenvalues of PC_1_, representing the highest proportion of the data variance, were higher in Intermediate (M = 2.59; SD = 0.22) in contrast to the Novice group (M = 2.30; SD = 0.30) (U = 18; *p* = 0.04; d = 1.10).

## 4. Discussion

This study aimed to examine the cardiorespiratory coordination of collegiate rowers under the prism of the network physiology of exercise. Results from this study showed that Intermediate rowers exhibited higher degree of cardiorespiratory coordination (CRC) compared to Novice rowers. More specifically, only 77% of the Intermediate group showed high CRC (i.e., PC_1_), while only 33% of the Novice group presented 1 single principal component analysis. The formation of an additional PC (i.e., PC_2_) in Novice rowers was the result of the shift of FeO_2_ or, alternatively, HR and VE, from the PC_1_ cluster of variables. The formation of an additional PC in Novice rowers was the result of the shift of FeCO_2_ from the PC_1_ cluster of variables.

Results of this study demonstrate the challenge around the VO_2max_ concept and its relevance to determine performance. When we examined the CRC among rowers within each group, we identified a member of the Intermediate group that had less degree of CRC despite the fact that she is an Olympian with the 2nd highest VO_2max_. Based on the applied grouping (i.e., Novice vs. Intermediate) of the initial rowing power and subsequent VO_2max_ rowing performance, it was expected that all of the Intermediate rowers would have had higher degree of CRC and vice versa for the Novice’s group. We showed that categorization of rowers just based on their respective VO_2max_ misclassified them and NPE framework provided more insights on rowers’ performance. Taking as reference the Olympian rower, where her VO_2max_ was the 2nd maximum, 54.2 mL/kg/min compared to 54.4 mL/kg/min which was the maximum value of a Novice rower, a reductionist approach would have implied that this athlete has reached the peak of her performance. However, with the NPE approach we can infer that her cardiorespiratory coordination has ample room for improvement, since her FeCO_2_ had decreased eigenvalues [52].

During maximum exercise FeCO_2_ and FeO_2_ present different patterns that accounts for 96–98% of variability in VO_2max_ [89], with FeO_2_ to be increased and FeCO_2_ at exhaustion compared to the beginning of the exercise [90]. At exhaustion, FeCO_2_ decreases driven primarily by the hyperventilation due to an increase in blood pH. The influence on FeCO_2_ on forming PC_1_ was shown elsewhere [52,63,64,65,67], therefore, since at maximum levels PC_1_ was loaded primarily by the FeCO_2_, this can be interpreted as improvement of individuals’ CRC in respect to buffering system and greater efficiency of the gas exchange system [91]. The reduction of CRC and the presence of PC_1_ driven by the FeCO_2_ may be related to the cardiorespiratory coupling and neuroautonomic regulation [92,93]. Higher intensities increase sympathetic activation as a result to feedback mechanisms from the pulmonary mechanoreceptors and peripheral chemoreceptors in response to carbon dioxide and oxygen levels during muscle perfusion [94]. Moreover, it has been suggested that the inability of the cardiorespiratory system to control and regulate its function is due to circular causality between VE, FeCO_2_, FeO_2_, and HR [95]. In addition, the importance of forming a PC_1_ due to FeCO_2_ and FeO_2_ in respect to CRC was also documented in runners under normoxia condition [96].

Physiological interpretation of the VO_2max_ needs to identify the limiting steps/factors that affect VO_2max_ and may change with training [97]. For example, in general limiting factor is the oxygen delivery (i.e., pulmonary ventilation and gas exchange, cardiac output, muscle blood flow, arterial oxygen content, muscle diffusion capacity or mitochondrial capacity) [13,98]. Having as reference point again the Olympian rower, lack of improvement on her already compared to the rest of the team maximum VO_2max_ value with team’s training cannot imply that the training program has failed or that it has elicited no adaptations. In elite athletes we expect to see an increase in peak cardiac output, in mitochondrial oxidative capacity, and in peripheral adaptations regardless any significant changes in the VO_2max_ [99,100]. This phenomenon was supported by our data that showed that Olympian’s CRC was not the highest (i.e., PC_2_), even though she had the 2nd highest VO_2max_. The increase in her number of PCs may also imply the absence of non-liner reconfiguration due to possible training adaptations [52] and less efficient CRC (i.e., increase in number of PCs and or decrease in PC eigenvalues) [52,68]. On the opposite side, the Novice athletes that had low or high VO_2max_ and a dimension reduction in their PCs that would imply that irrespective of their VO_2max_ they possess high coordinative structures [68,101]. In any of the aforementioned cases, a high or low CRC would reflect synergistic or non-synergistic adaptations of the involved systems under their respective environmental and/or systems-organs limitations [102]. Following the same line of thought, coaches may use this framework to take decisions about the potential of an athlete that demonstrates both high VO_2max_ and CRC values. In this case, it is logical to assume that this athlete has reached his/hers maximum athletic potential.

From a practical point of view, we showed that CRC can provide an insightful interpretation of a CPET. It seems that CRC might be more sensitive than VO_2max_ to assess the synergy between the cardiorespiratory system, the involved variability [103] between maximum presented efforts [104] that reflect the adaptive or not properties of the physiological network in response to VO_2max_ testing [53]. Current CRC approach via PCA evaluation of common CPET variables may be an alternative and supplementing assessment to the traditional CPET, as previous research has shown CRC is very responsive to cases related to maximum performance after training [52,67], after different fatigue states [64], and nutritional interventions [63].

Our results are not free from any methodological limitations. First, our sample size is limited to the number of the actual rowers that compete at the NCAA Division II and data were collected as part of team’s rowing performance testing. Second, no verification of the results of CPET was performed to determine whether it was a “true” VO_2max_ test nor any evaluation was made based on the established criteria [105,106]. In addition, since we did not measure lactate levels nor graphically examined the ventilatory threshold, it is possible that not all of the examined rowers gave their absolute maximum effort. This measurement was part of rowers’ annual evaluation, therefore we believe that the obtained results reflect rowers’ maximum efforts. No dietary control was performed, so our results may influenced by a diet high in carbohydrates or fat that subsequently would affect the metabolic rate with higher CO_2_ output at any given O_2_ uptake compared to a diet high in fat [107]. Another limitation of this study may be due to the applied discontinuous incremental rowing protocol to exhaustion, as continuous time series of the involved physiological variables were recorded and analyzed, but in a discontinued way. In order to capture interactions among physiological systems with PCA, a continuous time series analysis using a continuous CPET has been proposed [57]. This study examined the NPE framework in rowing using the PCA method, and for field applicability was decided to use a discontinuous incremental protocol instead of a continuous one. At the same time though, this is study’s major strengths as we showed that PCA can be used for discontinuous protocols too and its results can be immediately transferred from the laboratory settings to the on-water training and performance.

As the concept of CRC is not still wide known, future research need to utilize alternative to linear PCA modeling, such as non-linear PCA [108] or network component analysis [92,109,110]. Doing as such, a better understanding of the number of PCs involved to sports will be established and whether or not their increase or decrease is linked better physiological adaptation and performance. Moreover, future studies when perform CPET need to analyze the results under the NPE and CRC perspective in order to develop normative data across different sports, ages, gender.

## 5. Conclusions

A higher degree of CRC was displayed for the Intermediate rowers compared to Novice ones. Intermediate rowers were able to better manage the cardiometabolic by-products during the CPET possible due to enhanced bicarbonate buffering efficiency, as they relied less on ventilation. In accordance with the emerging CRC literature, we showed that CRC may provide insightful information to coaches, athletes and other stakeholders other than what a traditional CPET has to offer.

## Figures and Tables

**Figure 1 ijerph-19-13250-f001:**
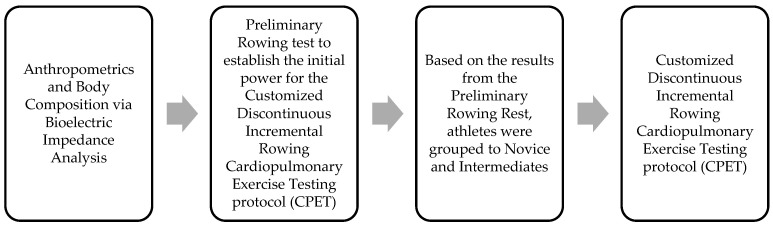
Research design.

**Table 1 ijerph-19-13250-t001:** Demographics.

	Age	BW (kg)	Height (cm)	BF (%)
N	18	18	18	18
Mean	20.17	70.89	170.17	22.01
Standard deviation	2.28	16.54	6.97	5.85
Minimum	18	47.50	159.50	12.30
Maximum	25	117.90	183.00	34.10

**Table 2 ijerph-19-13250-t002:** Projection Variables in PC_1_.

Novice Rowers							Intermediate Rowers						
ID	VO_2max (mL/kg/min)_	VE	HR	FeO_2_	FeCO_2_	#PC	ID	VO_2max (mL/kg/min)_	VE	HR	FeO_2_	FeCO_2_	#PC
1	45.1	0.87	0.93	0.78	0.91	1	10	32.6	0.79	0.80	0.77	0.90	1
2	45.1	0.17	0.01	0.97	0.95	2	11	50.2	0.89	0.48	0.82	0.96	1
3	41	0.69	0.87	0.78	0.90	1	12	38.5	0.94	0.89	0.71	0.08	2
4	40.8	0.96	0.92	0.03	0.63	2	13	45.7	0.78	0.67	0.80	0.94	1
5	34.9	0.07	0.08	0.97	0.96	2	14	46.4	0.86	0.87	0.64	0.91	1
6	54.4	0.92	0.89	0.13	0.50	2	15	48.3	0.83	0.88	0.61	0.95	1
7	25.2	0.94	0.89	0.15	0.46	2	16	47.4	0.84	0.50	0.90	0.92	1
8	41	0.88	0.03	0.96	0.98	2	17	41.8	0.84	0.90	0.69	0.91	1
9	43.3	0.72	0.68	0.87	0.93	1	18	54.2	0.93	0.94	0.17	0.48	2
Mean		0.69	0.59	0.63	0.80		Mean		0.86	0.77	0.68	0.78	
SD		0.34	0.42	0.40	0.21		SD		0.06	0.18	0.21	0.30	

Means and standard deviations are presented for each column and respective examined variable. ID: Rower’s assigned identification number; #PC: Indicates number of principal components; VE: Ventilation; HR: Heart rate; FeO_2_: Fraction of expired oxygen; FeCO_2_: Fraction of expired carbon dioxide. VO_2max_: Maximum oxygen consumption in mL/kg/min. ID#18 = Olympian athlete.

## Data Availability

Data are available upon request.

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
