# Peer review of "Cardiorespiratory Coordination in Collegiate Rowing: A Network Approach to Cardiorespiratory Exercise Testing"

_ijerph, 2022, doi:10.3390/ijerph192013250_

Round 1
Reviewer 1 Report
I want to thank the authors for their contribution in this field of research. However, there are a lot of issues in manuscript that need to answer it and correct before the publication.
Abstract
General comment: it is a need to provide a new version of your abstract. It is wordy for the reviewed. Furthermore, it should be clearly mentioned the part of methodology and results by providing scientific and significant numbers of your main variables.
Specific comment 1: L16-18: please remove this sentence.it is wordy.
Introduction
General comment: please insert the main hypothesis below the study purpose in the last paragraph.
Specific comment: L109-110: please remove this sentence.
Materials and methods
Specific comment 1: paragraph 2.1 please insert and construct a table with anthropometric characteristics of your participants.
Specific comment 2: paragraph 2.1: please construct a figure that illustrates clearly the study design.
Specific comment 3: L150-151: remove this sentence. It is not appropriate to justify why you decide to follow this testing protocol. If you want to provide it, please mention it in introduction.
Specific comment 4: Paragraph 2.3.3: please explain why you decide to follow these intensities.
Results
General comment: the chapter of results is "poor" and wordy.
Specific comment 1: please insert the units of VO2 max and for the other variables in table 1.
Specific comment 2: please construct different paragraphs and tables for each variable.
Discussion
Specific comment 1: insert the study limitations.
Specific comment 2: did you justify or not the hypothesis? Please mention it.
Conclusion
General comment: what is the rationale of your study? What is the "home message"? Please mention it.in conclusion.
Author Response
Dear Reviewer #1, please see below how we addressed your valuable feedback.
I want to thank the authors for their contribution in this field of research. However, there are a lot of issues in manuscript that need to answer it and correct before the publication.
-Thank you for your valuable feedback
Abstract
General comment: it is a need to provide a new version of your abstract. It is wordy for the reviewed. Furthermore, it should be clearly mentioned the part of methodology and results by providing scientific and significant numbers of your main variables.
-Thank you for your valuable comment. But keep in mind that the journal allows only 200 words for the abstract. Moreover, the methodology is listed, see lines 15-20. Results are listed in lines 21-23 and since this a PC analysis the traditional way of reporting the results is not the same. We hope you understand our point.
Specific comment 1: L16-18: please remove this sentence.it is wordy.
-Thank you for your valuable comment. This sentence is important for the abstract as shoes the purpose and hypothesis. Therefore, we cannot remove it. We hope you understand.
Introduction
General comment: please insert the main hypothesis below the study purpose in” the last paragraph.
-Thank you for your valuable comment. The hypothesis exists in lines 106-107 - We hypothesized that Intermediate rowers compared to Novice rowers will have both higher VO2max and CRC values.”
Specific comment: L109-110: please remove this sentence.
-Thank you for your valuable comment. This sentence relates to the significance of the study; therefore we believe that needs to be there so the reader gets more intrigued and keep reading the manuscript.
Materials and methods
Specific comment 1: paragraph 2.1 please insert and construct a table with anthropometric characteristics of your participants.
-Thank you for your valuable comment. Table 1 was created and placed in the Results section
Specific comment 2: paragraph 2.1: please construct a figure that illustrates clearly the study design.
-Thank you for your valuable comment. Figure 1 was created and placed in the methods
Specific comment 3: L150-151: remove this sentence. It is not appropriate to justify why you decide to follow this testing protocol. If you want to provide it, please mention it in introduction.
-Thank you for your valuable comment. There are cases like this, that in the Methods authors need to justify not just the how, but also the why they selected a specific protocol, especially when the protocol is not part of the research question. Therefore, for future studies that want to replicate our findings, this needs to be reported in here. We hope you understand.
Specific comment 4: Paragraph 2.3.3: please explain why you decide to follow these intensities.
-Thank you for your valuable comment. This is exactly what I just stated earlier. As we stated earlier, lines 154-155, there is no clear consensus in the sport of rowing on what protocols to use. Therefore, we used the current field practices line 157-161. Based on that info and references 77, 78 we applied this customized intensities.
Results
General comment: the chapter of results is "poor" and wordy.
-Thank you for your valuable comment. We respectfully disagree with being wordy. Actually it is only one paragraph that provides results that do not repeat in text and in table.
Specific comment 1: please insert the units of VO2 max and for the other variables in table 1.
-Thank you for your valuable comment. We added only for the VO2max, as the rest values are Eigenvalues
Specific comment 2: please construct different paragraphs and tables for each variable.
-Thank you for your valuable comment. This is not feasible, as there is not enough information to be presented in different and paragraphs-tables. This is how results are reported when PCA is presented. We hope that you understand.
Discussion
Specific comment 1: insert the study limitations.
-Thank you for your valuable comment. We have already the limitations presented, please see lines 294-312
Specific comment 2: did you justify or not the hypothesis? Please mention it.
-Thank you for your valuable comment. Line 106-107, we state that “We hypothesized that Intermediate rowers compared to Novice rowers will have higher CRC values”. In Discussion first paragraph we say line 225-227 “ Results from this study showed that Intermediate rowers exhibited higher degree of cardiorespiratory coordination (CRC) compared to Novice rowers. We believe that this sentence justifies the hypothesis.
Conclusion
General comment: what is the rationale of your study? What is the "home message"? Please mention it.in conclusion.
-Thank you for your valuable comment. We provided the rationale and significance of the this study in the last sentence of the Intro – see line 109 -110. We also provide evidence in the Intro about the problem of practice – see line 97-104. The take home message, as we state in the lines 324-326, is that coaches, athletes, other stakeholders may not only use the traditional CPET info to draw conclusions, but also to use a CRC to get more “insightful information” line 325
Reviewer 2 Report
Comments to the Authors:
General Comments:
The manuscript is well written and provides data pointing to the utility of the CRC methodology of assessing CRC during CPET. At this time I have only minor concerns about regarding the manuscript.
Throughout the text, there are small typographical and grammatical errors that should be addressed. For instance line 62 “Moreover, even though is difficult…”appears to be missing a word.
Introduction:
Line 83: While mentioned that an increase in PC eigenvalues and/or decrease in PCs is a represents a higher level of CRC, the cited reference is in a population of physically active males during cycling. As cardiorespiratory associated values and sex can be highly influenced on the task, is there any other evidence that these changes hold true in rowing and or treadmill CPET testing? While I feel that you have adequately addressed the gap in the literature and the need for more sensitive testing, this information could make for a stronger case that this worked needs to be done in other exercise modalities as well as across training status and sex.
Methods:
As an researcher that works primarily with competitive athletes and practitioner driven study designs I fully understand the use of the 250 W threshold used. I believe that the details to why this cutoff value is used should be addressed in the manuscript either in the methods or discussion as that threshold misclassified several individuals as novice and intermediate. In other words where did the 250 come from and why would a threshold such as 275 or 300 W not do a better job classifying individuals.
RESULTS:
What are the variables included in each PC? Table 1 shows PC1 but you mentioned how individuals also have a second PC so which variables are included in that?
Author Response
Dear Reviewer #2, please see below how we addressed your feedback.
General Comments:
The manuscript is well written and provides data pointing to the utility of the CRC methodology of assessing CRC during CPET. At this time I have only minor concerns about regarding the manuscript.
-Thank you for your valuable comment. It is much appreciated.
Throughout the text, there are small typographical and grammatical errors that should be addressed. For instance line 62 “Moreover, even though is difficult…”appears to be missing a word.
-Thank you for your valuable comment. We do not think that “appears” needs to be included.
Introduction:
Line 83: While mentioned that an increase in PC eigenvalues and/or decrease in PCs is a represents a higher level of CRC, the cited reference is in a population of physically active males during cycling. As cardiorespiratory associated values and sex can be highly influenced on the task, is there any other evidence that these changes hold true in rowing and or treadmill CPET testing? While I feel that you have adequately addressed the gap in the literature and the need for more sensitive testing, this information could make for a stronger case that this worked needs to be done in other exercise modalities as well as across training status and sex.
-Thank you for your valuable comment. As you probably noticed, the CRC and PC analysis along with the some other statistical approaches are novel approaches that we and our group examines in different populations. At the moment there is none research performed in sports like rowing, and this is a pioneer work. Therefore, there are no other works to be cited or referenced. As for the point regarding the work that needs to be done, we think that we addressed that in the Discussion, line 317-319, where we say that NPE and CRC needs to be tested across different sports, ages, gender. This is a truly pioneer work that we hope other researchers will get involved in NPE
Methods:
As an researcher that works primarily with competitive athletes and practitioner driven study designs I fully understand the use of the 250 W threshold used. I believe that the details to why this cutoff value is used should be addressed in the manuscript either in the methods or discussion as that threshold misclassified several individuals as novice and intermediate. In other words where did the 250 come from and why would a threshold such as 275 or 300 W not do a better job classifying individuals.
-Thank you for your valuable comment. This is very insightful comment and we have addressed why in the Methods section, 2.3.2. It is a field-based approach, coaches are using ~250 and on previous studies that they lack consensus 26, 36, 41, 77, 78, 81, 82. Rowing studies are limited, we even reached out to the USA Rowing to get what kind of protocols they use, we had already feedback from our Olympian – Paraguay Team. We do not disagree that a 275 or 300 may do or not a better job on classifying the athletes, but coaches and in particular this team’s coach is using the 250 as a criterion. We hope that this makes it clear for you.
RESULTS:
What are the variables included in each PC? Table 1 shows PC1 but you mentioned how individuals also have a second PC so which variables are included in that?
-Thank you for your valuable comment. As indicated in lines 211-241, the formation of PC2 in Novice rowers was the result of the shift of FeO2 or, alternatively, HR and VE, from the PC1 cluster of variables. As shown in Table 2, FeO2, VE and HR for Novice rowers showed the lowest projections onto PC1. For instance, in Participant 2, PC1 was composed of Feo2 and FeCo2 (highest projections) and PC2 was formed by HR and VE (lowest projections into PC1).
Reviewer 3 Report
In this manuscript, the authors conducted a cross-sectional study in which they enrolled 18 female collegiate rowing players who were classified as novice/intermediate and accepted cardiopulmonary exercise test, and cardiorespiratory coordination (CRC) method was used to analyze the co-variation among the cardiorespiratory variables. The idea of using CRC to test the cardiorespiratory coordination by the number of principal components is pretty new. I listed several major concerns need to be addressed.
1. Based on what criteria were the collegiate rowing players divided into novice/intermediate? Please support some solid standards.
2. Only 18 subjects were enrolled into this research. The sample size is too small. And this CRC method was applied to only one sport (collegiate rowing). All these limit the significance of this study. Please mention this in the limitation part
3. The authors alternatively use “higher degree of CRC/ low PCs / high PC eigenvalues” to say a same thing: higher level of cardiorespiratory efficiency-coordination. This will confuse the readers.
4. Please add more data or analysis results in table or figures.
Author Response
Dear Reviewer #3, please see below how we addressed your feedback.
In this manuscript, the authors conducted a cross-sectional study in which they enrolled 18 female collegiate rowing players who were classified as novice/intermediate and accepted cardiopulmonary exercise test, and cardiorespiratory coordination (CRC) method was used to analyze the co-variation among the cardiorespiratory variables. The idea of using CRC to test the cardiorespiratory coordination by the number of principal components is pretty new. I listed several major concerns need to be addressed.
- Based on what criteria were the collegiate rowing players divided into novice/intermediate? Please support some solid standards.
-Thank you for your valuable comment. As you correct stated, the idea of NPE and CRC is pretty new. Moreover rowing as sport is not evaluated (there is no so much interest and money involved) but we decided to focus on that just because nobody has done it so far. The justification on why and how we divided the rowers into Novice and Intermediates comes from the actual coach and his criteria of 250 W. Moreover as we have in the Methods section 2.3.2 and lines 154-163, there is no clear consensus, and also it is a field practice. We hope that this clarifies the concept.
- Only 18 subjects were enrolled into this research. The sample size is too small. And this CRC method was applied to only one sport (collegiate rowing). All these limit the significance of this study. Please mention this in the limitation part
-Thank you for your valuable comment. In the Methods section we explain that this is a “convenience” sample, because this is s routine test that the Rowing team is performing. Therefore, the number of 18 is fixed as this is the number of team members that compete. As for the CRC, yes, it is applied only on rowing, as this was the aim of the study. The idea of NPE and CRC is novel, so there is no other sports that were examined under this perspective, see line 319-321 where we say that CRC and NPE needs to be evaluated across different sports, ages, genders. We hope this effort to lead other researchers to expand our concept to new sports, see line 294-319. Anyway, we added more info make to highlight your concerns.
- The authors alternatively use “higher degree of CRC/ low PCs / high PC eigenvalues” to say a same thing: higher level of cardiorespiratory efficiency-coordination. This will confuse the readers.
-Thank you for your valuable comment. We use this depending on the section and the point that we want to make. We do understand that this may be confusing for some, but we believe that is necessary, as PCA statistical analysis is not that common, and readers have limited knowledge on how to evaluate the PCA outcomes. Therefore, it is necessary to go back and forth, since the concept is not as easy as a difference between two means.
- Please add more data or analysis results in table or figures.
-Thank you for your valuable comment. We added a figure to show the design and a descriptive table. Regarding the PCA table, this is the current way of reporting results, there are no other results to be reported. We hope that you understand the current statistical practices.
Round 2
Reviewer 1 Report
Thank you for the changes that made.
I suggest the current manuscript form for publication.
Reviewer 3 Report
In this manuscript, the authors conducted a cross-sectional study in which they enrolled 18 female collegiate rowing players who were classified as novice/intermediate and accepted cardiopulmonary exercise test, and cardiorespiratory coordination (CRC) method was used to analyze the co-variation among cardiorespiratory variables. The idea of using CRC to test the cardiorespiratory coordination by the number of principal components is pretty new. The authors responded to my questions and made some revisions. After explanations, the conclusion is less vulnerable. Overall, the manuscript is well written in English and can attract readers’ attention for its novelty.